# Genetic Screen in Adult Drosophila Reveals That dCBP Depletion in Glial Cells Mitigates Huntington Disease Pathology through a Foxo-Dependent Pathway

**DOI:** 10.3390/ijms22083884

**Published:** 2021-04-09

**Authors:** Elodie Martin, Raheleh Heidari, Véronique Monnier, Hervé Tricoire

**Affiliations:** Unité de Biologie Fonctionnelle et Adaptative (BFA), Université de Paris-CNRS, UMR8251 4 rue Marie Andrée Lagroua Weill Halle, CEDEX 13, 75205 Paris, France; elodie.martin@u-paris.fr (E.M.); raheleh.heidari_feidt@roche.com (R.H.); veronique.monnier@u-paris.fr (V.M.)

**Keywords:** Huntington’s disease, Drosophila, CBP, Foxo, Wnt signaling

## Abstract

Huntington’s disease (HD) is a progressive and fatal autosomal dominant neurodegenerative disease caused by a CAG repeat expansion in the first exon of the huntingtin gene (*HTT*). In spite of considerable efforts, there is currently no treatment to stop or delay the disease. Although *HTT* is expressed ubiquitously, most of our knowledge has been obtained on neurons. More recently, the impact of mutant huntingtin (mHTT) on other cell types, including glial cells, has received growing interest. It is currently unclear whether new pathological pathways could be identified in these cells compared to neurons. To address this question, we performed an in vivo screen for modifiers of mutant huntingtin (HTT-548-128Q) induced pathology in Drosophila adult glial cells and identified several putative therapeutic targets. Among them, we discovered that partial nej/dCBP depletion in these cells was protective, as revealed by strongly increased lifespan and restored locomotor activity. Thus, dCBP promotes the HD pathology in glial cells, in contrast to previous opposite findings in neurons. Further investigations implicated the transcriptional activator Foxo as a critical downstream player in this glial protective pathway. Our data suggest that combinatorial approaches combined to specific tissue targeting may be required to uncover efficient therapies in HD.

## 1. Introduction

Huntington’s disease (HD) is a progressive and fatal autosomal dominant neurodegenerative disease characterized clinically by motor, cognitive, and psychiatric deficits. It is a rare disease with overall prevalence of 2–6 persons per 100,000, depending on the geographic area [1]. HD is caused by a CAG repeat expansion in exon 1 of the gene encoding the large (3144 a.a.) huntingtin (HTT) protein. When the number of trinucleotide repeats exceeds 40 in the mutated *HTT* gene (*mHTT*), the disease is fully penetrant with an age of onset around 45 years, depending on the length of the repeat (reviewed in [2]). The more striking feature of the disease is prominent neurodegeneration in striatal medium spiny neurons and cortical pyramidal neurons, but, since *HTT* is expressed throughout the body, HD is also associated to peripheral dysfunctions notably in muscle, gut, and liver [3,4,5]. To date, no clinical trial has demonstrated treatment efficacy to slow down the disease [6,7].

Identification of the causative mutation in the *HTT* gene led to the development of a plethora of HD models from yeast to mammals. Among mammalian models, a large number of mice models have been generated that allowed functional studies in different genetic conditions. These include transgenic mHTT models, knock-in models, and transgenic YAC and BAC mice (reviewed in [8,9]). A large number of Drosophila models were also developed, taking advantage of the sophisticated genetic tools available in this organism and its short lifespan [10]. Drosophila models reproduce many features of the human pathology, including CAG repeat length dependence of neurodegeneration, and have also been used to investigate HD in peripheral tissues [11,12,13]. Using mammalian and invertebrate models, some key physiopathological mechanisms of HD have been identified, including altered neurotransmitter release and receptor activity, Ca^2+^ signaling impairments, mitochondrial dysfunctions, transcription dysregulation, and axonal transport impairments (reviewed in [14,15]).

Considering the prominent striatal neurodegeneration in HD, many of the initial investigations using these in vivo or cellular models expressing mHTT have been focused on neuronal impairments. In the past few years, more efforts have been dedicated to understand how other brain cells and notably astrocytes contribute to HD [16]. Indeed, astrocytes are critical for proper brain function since they associate with pre and post-synaptic compartments of the synapse, modulate synaptic neurotransmitter concentrations and provide the neuron with energy and substrates for neurotransmission. In addition, reactive astrocytes may participate to dysfunction in HD brain, although several studies suggest that this may represent a late event in the disease [17,18]. Interestingly, a recent study using a conditional mHTT-expressing BAC HD mouse model demonstrates that astrocyte-specific depletion of mHTT significantly rescues motor, psychiatric-like, neuropathological, and electrophysiological phenotypes in mice [19]. This strongly suggests that astrocytes are important contributors to the progression of the deleterious phenotypes observed in HD. Other types of glial cells, such as oligodendrocytes, are also believed to play an important role in the pathology [16]. However, it is currently not known whether similar pathways are affected in glial cells and neurons, and whether HD genetic modifiers identified in neurons could behave similarly in glial cells. It is thus important to improve our knowledge on these questions since discrepancies between these different brain cells may, in part, explain the poor success of clinical trials in HD.

To address this issue, we used an in vivo inducible glial model of HD and screened a set of genes, some of which having been previously related to HD by genetic or genomic studies. We uncovered several potent modifiers of HD pathology in glial cells, expanding the number of potential therapeutic targets. Unexpectedly, while some modifiers act in the same direction in neuronal and glial tissues, we discovered that depletion of dCBP in glial cells strongly improved the phenotypes of mHTT expressing animals in these cells, in contrast to previous findings obtained in neuronal tissue [20]. Investigating several putative targets of the CBP acetyltransferase, we found that this phenotypic improvement was lost in Foxo null animals, suggesting that dCBP repress Foxo in mHTT expressing glial cells and that dCBP depletion relieves this repression and contributes to HD mitigation. In agreement with this hypothesis, we showed that genetic interventions in the insulin pathway leading to increased Foxo signaling in the glial cells were also protective.

## 2. Results

### 2.1. Genetic Modifiers Can Be Identified in an Adult Glial Inducible Model of HD

A few previous studies performed in Drosophila looked at mechanisms of toxicity of mHTT in glial cells [21,22,23,24,25,26]. However, these studies were not designed to discriminate between developmental and adult-specific defects induced by mHTT. Here, we searched whether we could modify the course of HD in flies by genetic interventions exclusively in adult animals where mHTT is targeted specifically in glial cells. To this aim, we used a RU486-inducible GeneSwitch line under the control of the glial specific promoter repo (repoGS, [21]). The expression of mHTT in repoGS > UAS-HTT-548-128Q flies, induced by RU486 feeding from the first day of adulthood, led to a strong decrease of lifespan (Figure 1a). Induction of the repoGS alone did not impact lifespan in different genetic backgrounds (Appendix A). Compared to other studies using a classical repoGAL4 line, this indicates that mHTT expression in glial cells, specifically at the adult stage, is sufficient to induce toxicity, irrespective of developmental damages.

Next, we constructed a repoGS > UAS-HTT-548-128Q recombinant line and crossed it with a set of RNAi lines targeting genes that were selected for their potential role in HD. Some gain of function (GOF) lines were also used in the course of this study. A complete list of the genes tested with their mammalian orthologues is provided in Appendix A, while the description of the genotypes of the tested flies can be found in Appendix A. Most of the lines used in this study were validated in previous studies, as indicated in Appendix A. We compared the lifespan of the male flies resulting from these crosses to the one of control males issued from repoGS > UAS-HTT-548-128Q crossed with control lines adapted to the tested genotypes (see M&M for the choice of these lines). In the 27 independent longevity experiments performed in this study, we observed a highly reproducibility for the different control lines with a small dispersion around the mean lifespan (standard deviation <9%, Figure 1b). In contrast, a much larger dispersion was observed for the tested lines, indicating that HD glial modifiers can be identified in this way. Using a stringent threshold, we considered as positives the genetic conditions where the change of mean lifespan compared to the one of the control lines exceed 20% (corresponding to a Z-value of 2.3). With this threshold, we identified 24 enhancers and 22 suppressors (Appendix A). The full results of the screen are available in Appendix A and we discuss below some of the genes identified as modifiers of the mHTT longevity phenotype, with a special emphasis on conditions leading to extended lifespan.

### 2.2. Known Modulators of Neuronal polyQ-Induced Toxicity Exhibit Various Effects in HD Adult Glial Cells

We first assayed in our adult glial HD model the potency of a set of chaperones to rescue the pathology when overexpressed, as it is the case in several models of polyQ diseases targeting neuronal cells [27]. Overexpression of two chaperones, Hsc70Cb and HSP68, and of the DNAJ1 co-chaperone robustly increased the longevity of the HD flies while overexpression of the human HSPA4L chaperone was inefficient (Figure 2a). Similarly, like in neurons [28], loss of function of Rab5, which interacts indirectly with mHTT through HAP40, exacerbates the glial pathology while two independent overexpressing lines led to a mild 10–13% increase in lifespan (Figure 2a).

Reduction of HDAC1 has been shown to protect neuronal cells in various polyQ models, including an HD Drosophila model [29,30,31]. We found that RNAi inhibition in adult glial cells did not lead to a significant increase in lifespan (Figure 2b). Similarly, we did not find a protective effect after RNAi depletion of genes involved in the kynurenine pathway like the tryptophan 2, 3-dioxygenase (*v*/TDO) or the kynurenine 3-monooxygenase (*cn*/KMO) (Appendix A; Figure 2b), in spite of their involvement in neuronal toxicity in HD and SCA3 [32,33]. Similarly, several genes (*faf*, *Gbeta13*, *Synd*, *Taf4*, *Pgi*, *and Gapdh1*), encoding proteins that interact physically with mHTT and have been identified as modulators of toxicity in the fly eye neurons [34], were also unable to change the fly lifespan when inactivated or overexpressed in glial cells (Appendix A). Altogether, these results point to the existence of cell-specific modifiers of HD pathology.

### 2.3. Impact of the Modulation of Energy Production Pathways in HD Adult Glial Cells

One important difference between neurons and astrocytes is their different regulation of energy metabolic pathways that results in improved resistance to oxidative stress in astrocytes. We addressed the question of the sensitivity of glial cells to energy pathways perturbations by targeting several enzymes involved either in glycolysis (Pgi, Pfk, fbp, Gapdh1, and Pyk) or in the Krebs cycle (Mdh1, Nc73EF, SdhA, and SdhB). To our surprise, RNAi-induced depletion of many of these genes did not impact the longevity of the HD flies (Appendix A), suggesting that glial cells may develop some compensatory mechanisms. One exception was the pyruvate kinase gene (*Pyk*) for which expression of one RNAi line results in a strong lifespan increase (Pyk-RNAi1, +44%, in 4 independent experiments, Figure 3a). A lower but significant effect was also observed with a stronger RNAi line (Pyk-RNAi2) which in addition resulted in decreased lifespan when expressed with a repoGS line alone, as expected from [35]. No such decrease was observed with the Pyk-RNAi1 line (Appendix A). The human ortholog of Pyk, PKM, has been shown to interact physically with mHTT in several studies [34,36] and it is possible that the outcome of Pyk depletion in HD glia is highly sensitive to the level of this protein. This will require further investigations in the future.

In HD animals, depletion of succinate dehydrogenase has been observed by several groups [37,38]. We found that overexpression of the SdhB enzyme (but not SdhA), using a UAS transposon inserted upstream of the gene, conferred a weak protection (11% ± 3%) against glial HD induced toxicity (Appendix A), although this was below our stringent threshold.

Finally, we assayed the protection conferred by overexpression of *spargel (srl)*, the Drosophila orthologue of PGC1α, a major regulator of mitochondrial biogenesis. A previous study has shown that this treatment was efficient against mHTT-induced dysfunctions in a fly neuronal model [39]. Glial overexpression of *spargel* resulted here in a strong 50% increase of mean lifespan (Figure 3b). This confirms the interest of pharmaceutical strategies aimed at boosting PGC1α in HD which could be efficient in various brain cells [40].

### 2.4. HD Modifier Genes Involved in Calcium Signaling

As perturbations in calcium regulation and signaling have been well documented in HD neurons, we assayed the importance of some genes in these pathways in HD glia. Cells use several ways to modulate cytoplasmic Ca^2+^ levels, notably by using the inositol 1,4,5-trisphosphate (InsP3) to activate the inositol 1,4,5-trisphosphate receptor (IP3R) that releases Ca^2+^ from the endoplasmic reticulum (ER). Importantly, mutant HTT, ataxin-2 and ataxin-3 proteins specifically bind to the carboxy-terminal region of IP3R1 in rat neurons and increase its sensitivity to InsP3 [41,42]. In response to depletion of Ca^2+^ in the ER store, Orai and Stim activate a store-operated Ca^2+^ entry (SOCE) from the extracellular matrix across the plasma membrane, leading to a second rise in cytosolic Ca^2+^, which is then pumped back into the ER by the SERCA pump [43].

In our Drosophila adult glia HD model, RNAi-induced depletion of Itpr (the single IP3R in fly) did not rescue the fly lifespan significantly (Figure 4a). A RNAi line targeting Orai did not improve either the HD phenotype while additional crosses with two Orai mutants resulted in a 5% to 16% increase in lifespan. Moreover, RNAi-induced depletion of Stim, the major partner of Orai, did not result in phenotypic improvement (Appendix A). Therefore, we consider that our data do not provide evidence for a protection of HD glial cells by inhibition of SOCE in Drosophila.

In contrast to these negative results, we found that depletion of SERCA in glial cells conferred a strong protection to the HD flies with increases in lifespan ranging from 39% to 61% with two independent RNAi lines (Figure 4b). This suggests that modulation of the cytoplasmic Ca^2+^ levels in glial cells can be a therapeutic strategy, like in neurons.

### 2.5. Depletion of dCBP in Glial Cells Mitigates HD Pathology

Changes in cytoplasmic calcium concentrations may activate downstream pathways by activation of various kinases. This may results in the activation of the CREB transcription factor through interaction with co-activators like CBP/p300 and CRTCs (Reviewed in [44,45]). Calcium dependent phosphatases like calcineurins act as negative regulators of this pathway. We depleted by RNAi some members of this pathway and assayed the consequences of such depletions on HD toxicity in glial cells. We found no effect on fly survival of depletion of calcineurin subunits, the single Drosophila CRTC gene and CrebA (Appendix A). Depletion of CrebB significantly reduced the lifespan of HD flies but overexpression of this gene did not impact significantly this phenotype (Figure 5a). Strikingly, RNAi depletion in glial cells of the single Drosophila CBP/p300 ortholog dCBP, encoded by the *nejire* (*nej*) gene, increased lifespan by 80% (Figure 5b). A similar treatment in adult neurons was inefficient to improve mHTT-dependent decrease in lifespan (Figure 5c). Since this line has not been used in previous experiments, we checked that it efficiently reduced nej mRNA expression by 60–80% (Appendix A). Altogether, this suggests that depletion of dCBP in Drosophila adult glia may specifically rescue HD pathology in glial cells in a CREB independent manner.

To understand how dCBP depletion improves the HD lifespan phenotype, we first asked, by qPCR and Western blot analysis, whether it reduced the level of mHTT produced in our glial model. This was not the case neither for the mHTT mRNA nor the protein level (Figure 5d; Appendix A). Indeed, in both cases, we observed a trend towards increased levels when dCBP is depleted. This may result from a better survival of glial cells expressing mHTT, compared to cells containing wild-type levels of dCBP. Then, we assayed several available RNAi or shRNA dCBP lines to ensure that the protective effects cannot be ascribed to repression of an off-target gene. All the tested lines led to an increase of fly lifespan, ranging from 40% to 88% (Appendix A; Figure 6a). Overexpressing dCBP in glial cells did not change significantly the lifespan of HD flies (Appendix A; Figure 6a).

### 2.6. Depletion of dCBP in Glial Cells Improves Locomotor Defects in HD

Having established the robustness of the lifespan rescue effect, we asked whether dCBP depletion could improve other phenotypes. We measured the performance of flies in longitudinal geotaxis assays which are widely used in fly models of neurodegeneration.

Uninduced repoGS > UAS-HTT-548-128Q expressing a control RNAi construct exhibit a shallow decrease of performances by day 12 that is more pronounced by day 20. When the same flies were submitted to RU486 feeding from the first day of adulthood, they present strong locomotor defects by day 12 where their score is similar to the one of 20-days old uninduced flies. Four-days and eight-days old flies depleted for dCBP behaved similarly to control flies expressing a control RNAi, or to uninduced flies (where mHTT is not induced by RU486 feeding). However, at days 12 and 15, the performances of the dCBP depleted flies were much better than the one of control flies expressing a control RNAi (Figure 6b). At day 20, their performances remained close to those of uninduced flies. Thus, dCBP depletion in adult glial cells is sufficient to improve several key features of HD pathology in fly.

### 2.7. Glia Depletion of dCBP Rescues HD in a Foxo-Dependent Manner

In addition to its action on histone acetylation, CBP can modulate the activity of transcription factors like p53 or Foxo through direct acetylation [46,47]. It can also modulate the Wnt pathway in a complex manner through interactions with Beta-catenin and TCFs [48]. Interestingly, mHTT has been shown to interact with p53 [35] and to alter the binding of Beta-catenin in the destruction complex [49], while daf16/Foxo has been implicated in vivo in protection against HD in worms [50].

To assay the involvement of these different pathways in the protection conferred by dCBP in glial cells expressing mHTT, we took advantage of the existence of viable loss of function mutants of *p53* and *foxo* in fly. When we co-expressed mHTT and an RNAi directed against dCBP in glial cells of flies devoid of p53, we still observed an increase of their longevity compared to control flies (Figure 7a). Thus, p53 is dispensable for the dCBP-mediated protection in flies.

In contrast, when we performed the same experiment in flies devoid of Foxo, we observed a complete suppression of the protective effect conferred by dCBP depletion (Figure 7b). Thus, at least one copy of the *foxo* gene is required in this HD protective dCBP pathway. Moreover, we found that direct overexpression of *foxo* was able to rescue HD pathology in glial cells, as shown in Figure 7c, suggesting that dCBP depletion could act by increasing Foxo activity. Since transcriptional activity of Foxo is also under the control of the insulin pathway, we assayed this pathway and found that attenuation of insulin signaling by expression of a dominant negative insulin receptor (InR-DN) or overexpression of the phosphatase PTEN also increased the lifespan of flies expressing mHTT in glial cells (Figure 7c).

### 2.8. Identification of New HD Modifiers in the Wnt Pathway

A potential involvement of the Wnt pathway in HD has been previously observed by Dupont and collaborators [23], using a fly model in which a short human HTT fragment encompassing Exon 1 with 93 CAG repeats was expressed during development and adulthood in glial cells with the repoGAL4 driver. RepoGAL4 > UAS-Htt-Ex1-93Q flies heterozygous for either *arm/beta-catenin* or *pan/dTCF* or overexpressing the *Axin* gene exhibit longer lifespan and improved locomotor activity compared to their repoGAL4 > UAS-HTT-Ex1-93Q/+ controls [23]. This was also the case when the longer HTT-548-128Q fragment was expressed specifically in adult glial cells together with transgenes expressing a RNAi targeting arm or expressing a dominant negative dTCF (dTCF-DN) or the *Axin* gene (Figure 8a; Appendix A). This indicates that interventions in this pathway limited to adulthood are also efficient to improve the pathology. We extended the number of putative therapeutic targets by screening for genes located upstream or downstream of dTCF. We found that RNAi-mediated depletion of Gint3, the ortholog of UBXN6, a positive regulator of this pathway [51], was also sufficient to improve the lifespan of HD flies in our model (Figure 8b). This was also the case for depletion of *pygo* and *kto* (Figure 8b) which are members of the Wnt enhanceosome and the associated transcription complex [52]. In contrast, targeting *groucho* (the orthologue of the transcriptional repressor TLE in mammals) resulted in decreased lifespan.

## 3. Materials and Methods

### 3.1. Drosophila Lines and Culture Methods

The repoGS line was previously described in [53]. The UAS-HTT-548-128Q line was kindly provided by F. Maschat. The lines used for genetic screening are listed in Appendix A and were obtained from the Bloomington stock center or the Vienna Drosophila Research Center (VDRC), except lines related to the *drl* and *drl2* genes that were provided by JM. Dura. The fly food used for all experiments contained 82.5 mg/mL yeast, 34 mg/mL corn meal, 50 mg/mL sucrose, 11.5 mg/mL agar, and 27.8 μL/mL methyl 4-hydroxybenzoate (stock solution 200 g/L in ethanol).

### 3.2. Lifespan Experiments

Flies were collected within 24 h of eclosion under brief CO_2_ anesthesia, housed in groups of 30, and raised at 26 °C under a 12 h–12 h light-dark cycle. They were transferred every two days on fresh food, and dead flies were counted. For induction, RU486 was incorporated in the fly food from a 10 mg/mL stock solution in ethanol. All the longevity experiments were performed on male flies, except otherwise mentioned.

During the screen, we paid a particular attention to use adequate controls for each transgene (TG) analyzed. Thus, longevity of the flies with different genotypes (such as UAS-HTT-548-128/+; repoGS/TG) were compared to appropriate control flies: a UAS-mcd8-GFP was used for GOF constructs; a Sgs1[v21206] when TG corresponds to a VDRC line; TRiP.HMS02542 or TRiP.JF03252 lines when TG corresponds to a TRIP line. These control lines target genes that are not expressed in the glial tissue and thus do not interfere with the pathology. Their use prevented from detection of any false positive due to GAL4 dosage (due to the presence of 2 UAS sequences), non-specific effects linked to activation of RNAi pathways or different genetic backgrounds. Statistical significance of differences in survival curves was evaluated with logrank tests.

### 3.3. Quantification of Transcripts by qRT-PCR

RNA extractions were performed with the PicoPure^TM^ RNA Isolation Kit (Thermo Fisher Scientific, Waltham, MA, USA) and treated with dsDNase (Thermo Fisher Scientific) according to the manufacturer’s instructions. cDNAs were synthesized from isolated total RNA samples using SusperScript™ III Reverse Transcriptase (Thermo Fisher Scientific). qPCRs were performed with the qPCR Mix (Promega, Madison, WI, USA) on a LightCycler480 (Roche, Basel, Switzerland). The ribosomal gene *rp49* was used as an internal reference for normalization. The primers used for amplifications were for the *HTT* gene: 5′-TGCCAGCACTCAAGAAGGACAC-3′ and 5′-TGAGCAGCACGCCAAGAATCAG-3′; for the *nej/dCBP* gene: primers 1: 5′-AGGGCGATACGGTCACACT-3′ and 5′-TGAACTGATCCTTTTTGATTTGG-3′ and primers 2: 5′-GTGGGCACTCAGATGGGTATG-3′ and 5′-CATGCCTGGTATGGCGTTCA-3′. Quantifications were made on three to four independent biological samples for each biological sample. Statistical significance was assessed by unpaired *t*-test using Prism V6.01 software (GraphPad Software, San Diego, CA, USA).

### 3.4. Western Blot Analysis

Duplicate aliquots of 30 fly heads were lysed at 4 °C in 70 µL of KCl Buffer (50 mM Tris–HCl pH8, 10% Glycerol, 5 mM EDTA, 150 mM KCl) supplemented with 1% cOmplete protease inhibitor (Roche). Proteins were separated by SDS-PAGE on 4–12% polyacrylamide gradient gels and then electrotransferred onto nitrocellulose membranes (Schleicher & Schuell BioScience, Dassel, Germany). Standard immunochemistry protocols were used with primary mouse anti-huntingtin (MAB2166, 1/1000, Sigma-Aldrich, Saint-Louis, MO, USA) and mouse HRP secondary antibodies (1/20,000, OriGene Technologies, Rockville, MD, USA). The primary mouse anti-LaminC (LC28.26, 1/1000, DSHB, Iowa City, IA, USA) was used as a loading control. Immunoreactivity was imaged and quantified with Immobilon^TM^ Western Chemiluminescent HRP Substrate (Millipore) on an Amersham Imager 600 apparatus (Cytiva).

### 3.5. Negative Geotaxis Assays

Locomotor activity was assessed by negative geotaxis assays. Flies were placed in 5 tubes per genotype by groups of 10 and made to fall to the bottom by tapping the tubes five consecutive times. Movies were acquired using a HF12.5HA-1B Fujinon Camera. Five movies were made on the same samples of flies, separated by a rest period of 2 min. The ability to climb, given as a mean distance for each group of flies, was assessed at 7 s using the Fiji Software. Five independent experiments with such configuration were performed. Statistical analysis was performed with GraphPrism software by comparing each genotype to the control flies or to uninduced flies with an unpaired two sided *t*-test.

## 4. Conclusions

HD is a devastating neurodegenerative disorder and although the causal dominant mutation in the *HTT* gene was identified more than 20 years ago, there is still no efficient treatment. The difficulties to uncover such a treatment may come from different reasons: (1) the complexity of the HD pathology since the mHTT protein interacts with hundreds of proteins and disrupts many cellular processes [14]; (2) the ubiquitous expression of HTT, meaning that specific pathways may be differentially affected in different cells types; (3) the fact that HD is not only a degenerative disease but has also a neurodevelopmental component [54]. It is thus important to identify which processes are altered by mHTT in each of these temporal phases.

In this study, we searched for genetic modifiers of HD pathology in an in vivo Drosophila model. This model presents two main characteristics—it is dedicated to uncover mHTT induced pathological pathways specifically in glial cells which are under studied as compared to neurons—it specifically addresses pathological mechanisms occurring during adulthood, bypassing any neurodevelopmental effects that could interfere in these mechanisms. To these aims, we took advantage of a newly generated GeneSwitch glial specific line to express mHTT simultaneously to transgenes expressing RNAi or cDNA constructs for a set of 137 candidate genes and scored for changes in fly lifespan. From this screen, we uncovered 24 enhancer and 22 suppressor lines that are listed in Appendix A. It would have been desirable to complete this genetic analysis with HD transcriptome data in the context of the suppressor genes. However, such an analysis was prevented by the small proportion of the glial cells in the fly brain (10% of total), resulting in insufficient purification of glial specific mRNAs in our hands. We also did not address the question of the specificity of such suppressors in the different glial cell types, due to the lack of specific GeneSwitch drivers. These two aspects are important directions of research in the future.

Overall, our screen identified some genetic interventions that appeared to be efficient in brain cells in all situations, in spite of cell specificities (see below). This includes the protection provided by chaperone overexpression or the downregulation of the canonical Wnt signaling, as shown in this study and previous work. However, in many cases, it did not extend to the glial HD pathology the potency of several genes previously identified in screens for modification of mHTT induced photoreceptors degeneration in Drosophila. For instance, we did not observe significant changes in lifespan when we modulated the expression of p53 or members of the kynurenine pathway. This may reflect the fact that these genes modulate the pathology in some but not all the type of cells. Alternatively, it could result from different susceptibility to mHTT in the developmental phase (which has a major influence on eye phenotypes) and in adulthood. In both cases, it means that therapeutic interventions have to be evaluated at different stages and, when required, with improved targeting of the drugs.

The different behavior of modifier genes observed between neurons and glial cells in HD may result from their strikingly different energy metabolism [55]. Indeed, using standard repoGAL4 and elavGAL4 lines to express an HTT-Ex1-93Q toxic fragment, Besson et al. showed that the co-overexpression of the mitochondrial decoupling protein UCP2 in glial cells but not in fly neurons resulted in protection, as revealed by extended lifespan and improved behavior [22]. In this study, we investigated whether downregulation in glial cells of enzymes involved in glycolysis (including the rate limiting Pfk) may protect flies from mHTT toxicity, as it has been observed in neurons [56]. In most cases, this was not the case, emphasizing the difference between the two types of cells. However, surprisingly, partial depletion of pyruvate kinase (Pyk) in glia appeared to protect the HD flies. An interesting potent mechanism to be further explored would be that Pyk depletion contribute to downregulation of the Wnt pathway, a major toxic pathway upregulated in HD (see below), as observed for its mammalian orthologue PKM2 [57,58].

We also document in this study strikingly different responses to mHTT in neurons or in glial cells, as exemplified by modulation of genes controlling calcium homeostasis and the *dCBP* (*nej*) gene. In the first case, we did not find improvement in lifespan when SOCE related genes or Itpr, the IP3R1 orthologue, were downregulated in glial cells, in contrast to studies performed in neuronal cell models and mouse models [41,59,60]. Surprisingly, using two independent RNAi constructs, we found that inactivation of the SERCA pump, that is required to replenish the ER with Ca^2+^, resulted in a significant improvement of the HD glial pathology. This suggests that perturbations of Ca^2+^ homeostasis may be significantly different in HD neurons and glial cells, a hypothesis that should be further tested in mouse models.

The most striking case of HD-induced toxicity discrepancy between neurons and glial cells is illustrated by the *nejire* gene. To our surprise, we found that depletion of dCBP in adult glia alleviated the toxicity of mHTT in these cells, leading to increased lifespan and improved locomotor activity. Previously, in the framework of a HD fly eye model where a 127Q transgene was expressed, overexpression of dCBP was found to rescue photoreceptors degeneration [20]. A reason for these opposite results could come from the use of the eye as a read-out of toxicity. Indeed, eye development is exquisitely sensitive to loss or gain of function of dCBP [61] which complicates the interpretation of the data. Using the same line than in [20] (nej^EP1179^) to overexpress dCBP specifically in adult neurons of elavGS > UAS-HTT-548-128Q flies, we found that this intervention did not increase the short lifespan of these HD flies (data not shown). Depletion of the dHDAC1 (*rpd3*) deacetylase that counteracts dCBP activity, was also unable to rescue reduced fly lifespan in a non-inducible neuronal HD model although it has a positive effect on photoreceptor neurodegeneration during development [30], emphasizing again the differences of toxicity during development and adulthood in polyQ diseases. To our knowledge, no studies have been performed in mouse to investigate stage-dependent effects of CBP modulation on HD pathology, but they would be highly instructive in view of the fly data.

How does dCBP depletion in glial cells exert its protective effect in the HD fly model? CBP acetylates both histones and other client proteins and has a plethora of interactors. We focused on some well characterized CBP targets and assayed whether their modulation could improve the HD glial pathology in flies. We did not found evidence for a role of CREB proteins or p53 in the glial protection. In contrast, overexpression of the single *foxo* gene in flies, either directly or by playing with insulin pathway members, rescued significantly the decrease of fly lifespan. Foxo acetylation by CBP represses its transcriptional activity and this occurs in an oxidative stress (OS) dependent manner in the case of the mammalian Foxo4 [46]. Increased OS is associated to HD, which may result to sub-optimal Foxo activity. Interestingly, the rescue of HD phenotypes by dCBP depletion was fully suppressed in animals depleted for Foxo, supporting such a model. However, we cannot exclude that other targets could be involved in the protective effect of dCBP depletion in glial cells. Indeed, among the new suppressors of the pathology uncovered in this study, *pygo*, *kto,* and *skd* are good candidates to mediate CBP-dependent pathological effects. Pygopus is a member of the evolutionary conserved Wnt enhanceosome [62] that binds Legless (BCL9 in mammals), an interactor of arm/Beta-catenin, and both genes are required for Wingless target gene transcription. Importantly Legless can also bind CBP which may mediate this transcriptional activation [52], likely through the mediator complex subunits Med12 and Med13 (kto and skd) [63]. Thus, taking into account that WNT and FOXO pathways may also directly interact [64,65], additional work is still required to elucidate the mechanisms involved in CBP contribution to HD pathology in glial cells.

Altogether, our study emphasizes the need to take into account the cellular heterogeneity of HD pathological mechanisms that may counteract some therapeutic strategies based on single cell types. Combinatorial approaches combined to specific tissue targeting may thus help to uncovered efficient therapies in HD.

## Figures and Tables

**Figure 1 ijms-22-03884-f001:**
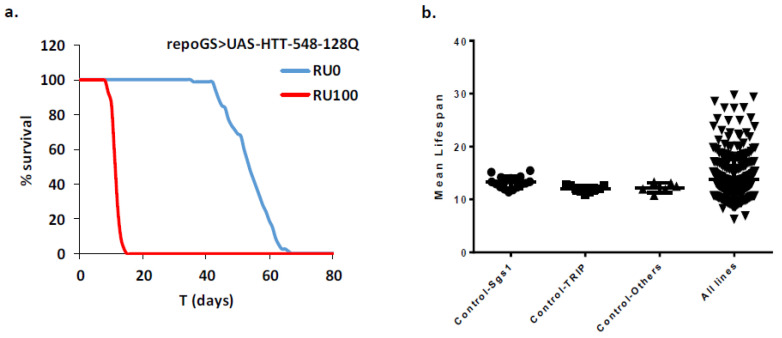
Identification of HD modifiers in Drosophila glial cells. (**a**) Progeny issued from a cross between the GeneSwitch repoGS line and a UAS-HTT-548-128Q transgenic line exhibit strongly reduced lifespan (mean lifespan (MLS) = 10.7, *n* = 205) when induced with 100 µg/mL of RU486 incorporated in the food at the emergence (RU100), compared to uninduced flies (RU0, MLS = 52.8, *n* = 80, pLogRank < 10^−10^). (**b**) A repoGS > UAS-HTT-548-128Q line was crossed with lines to be tested or to various control lines (see Appendix A for detailed genotypes) and the MLS of the progeny was scored. The distribution of the MLS was highly reproducible for the controls (MLS = 13.2, 12.0, 12.2; SD = 1.13, 0.66, 0.92; Nexp = 17, 9, 6, respectively, from left to right) while a large dispersion of the MLS in tested lines illustrates the identification of HD modifiers. With a threshold of >20% MLS fold change, 24 enhancers and 22 suppressors were identified.

**Figure 2 ijms-22-03884-f002:**
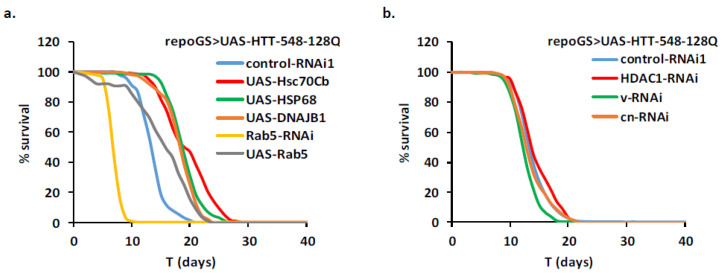
Analysis of known neuronal HD modifiers in glial cells. A repoGS > UAS-HTT-548-128Q line was crossed with transgenic lines allowing modulation of expression of known neuronal HD modifiers. The adult progeny (see Appendix A for detailed genotypes) was transferred at emergence on fly food containing RU486 (100 µg/mL) to induce the transgenes in glial cells and scored for their lifespan. (**a**) Overexpression of chaperones Hsc70Cb (UAS-Hsc70Cb), HSP68 (UAS-HSP68), the DNAJ1 co-chaperone (UAS-DNAJB1) and of Rab5 (UAS-Rab5) are protective against mHTT induced glial pathology as compared to control (MLS = 18.7, 18.3, 17.5, 14.5; pLogRank < 10^−10^, 10^−10^, 10^−10^, 10^−6^; *n* = 154, 148, 164, 76, respectively). In contrast, RNAi mediated depletion of Rab5 (Rab5-RNAi) is strongly deleterious (MLS = 6.3; pLogRank < 10^−10^; *n* = 60). Control-RNAi1 flies: MLS = 12.9, *n* = 122. (**b**) RNAi mediated depletion of HDAC1 (HDAC1-RNAi), cinnabar/KMO (*cn*/cn-RNAi) and vermilion/TDO (*v*/v-RNAi) have no significant effect or a slightly deleteriou-s effect on lifespan of glial HD flies (MLS = 13.2, 13.8, 11.9, 12.8; pLogRank > 10^−2^, 10^−2^, <10^−4^; *n* = 140, 142, 201, respectively). Control-RNAi1 flies: MLS = 13.2, *n* = 147.

**Figure 3 ijms-22-03884-f003:**
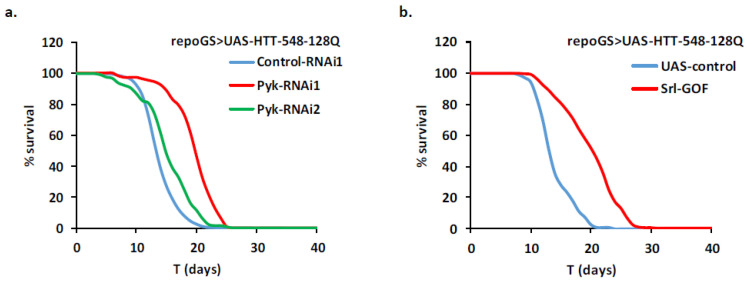
Identification of modifiers of mHTT-induced glial pathology involved in energy production. A repoGS > UAS-HTT-548-128Q line was crossed with transgenic lines modulating *Pyk* (Pyk-RNAi1 and Pyk-RNAi2) and *srl* (srl-GOF) genes (see Appendix A for detailed genotypes) and the progeny were scored for their lifespan after induction with RU100. (**a**) Adult glia depletion of Pyk with 2 independent RNAi lines results in increased lifespan of mHTT expressing flies (MLS = 18.7, 14.4; pLogRank < 10^−10^, 10^−5^; *n* = 106, 131, respectively). Control flies: MLS = 13.2, N = 147. (**b**) Overexpression of *srl* is also strongly protective, as shown by two independent experiments (MLS= 19.9, 18.2; pLogRank < 10^−10^, 10^−10^; *n* = 144, 136, respectively). Control flies: MLS = 13.3, *n* = 106.

**Figure 4 ijms-22-03884-f004:**
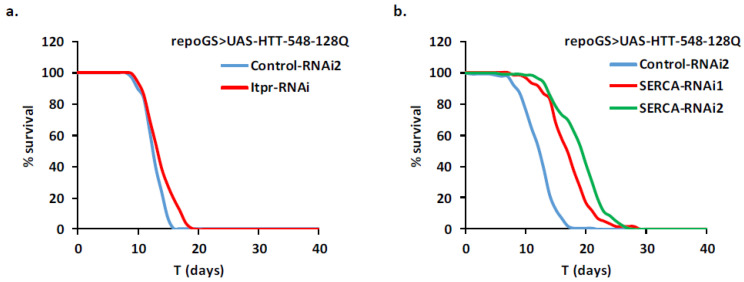
Repression of SERCA is protective in HD glia. A repoGS > UAS-HTT-548-128Q line was crossed with transgenic lines modulating Itpr (Itpr-RNAi) or SERCA (SERCA-RNAi1 and SERCA-RNAi2) (see Appendix A for detailed genotypes) and the lifespan of the progeny after induction with RU100 was compared. (**a**) Adult glia depletion of Itpr did not change the median lifespan of the mHTT expressing flies but marginally increased their mean lifespan (MLS = 13.1 vs. 12.0 for controls; *n* = 99, 150, respectively; pLogRank < 10^−4^). (**b**) Adult glia depletion of SERCA strongly increased the mean lifespan of the mHTT expressing flies (MLS = 16.4, 18.3; *n* = 59, 150; pLogRank < 10^−10^, respectively). Control-RNAi2 flies: MLS = 11.5, *n* = 141.

**Figure 5 ijms-22-03884-f005:**
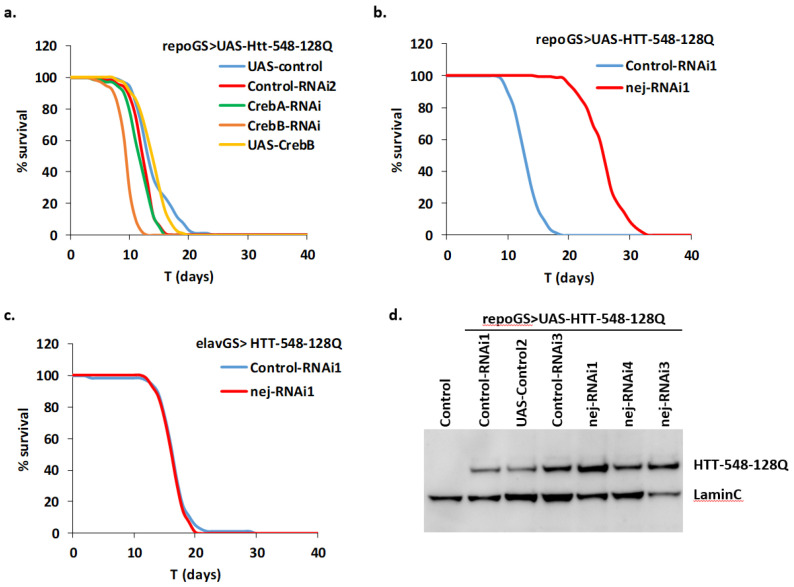
Repression of dCBP is protective in HD glia. (**a**,**b**) A repoGS > UAS-HTT-548-128Q line was crossed with transgenic lines modulating genes involved in the CREB pathway (see Appendix A for detailed genotypes) and the lifespan of the progeny after induction with RU100 was compared. (**a**) Adult glia depletion of CrebA (CrebA-RNAi) or overexpression of CrebB (UAS-CrebB) did not change the median lifespan of the mHTT expressing flies (MLS = 11.0, 13.1; pLogRank > 0.1; *n* = 74, 143, respectively). In contrast, repression of CrebB (CrebB-RNAi) was deleterious for the flies (MLS = 8.7; pLogRank < 10−10; *n* = 80). Control flies: MLS = 11.5, 13.3; *n* = 121, 106 for control-RNAi2 and UAS-control, respectively. (**b**) Adult glia depletion of nej/dCBP (nej-RNAi1) strongly increases the median lifespan of mHTT expressing flies (MLS = 25.0; pLogRank < 10−10; *n* = 86). Control flies: MLS = 12.4; *n* = 145. (**c**) Adult neuronal depletion of nej (nej-RNAi1) does not modify the reduced median lifespan of neuronal mHTT expressing flies (MLS = 15.8; pLogRank > 0.2; *n* = 72). Control flies: MLS = 15.6; *n* = 110. (**d**) The level of expression of huntingtin in repoGS > UAS-HTT-548-128Q flies co-expressing RNAi constructs targeting dCBP or control transgenes was analyzed by Western blot, with antibody directed against LaminC as a loading control. The specificity of the human HTT antibody was checked in a w^1118^ strain. Overall, the data do not indicate a reduced expression of mHTT after dCBP depletion. See Appendix A for detailed genotypes and Appendix A for quantification.

**Figure 6 ijms-22-03884-f006:**
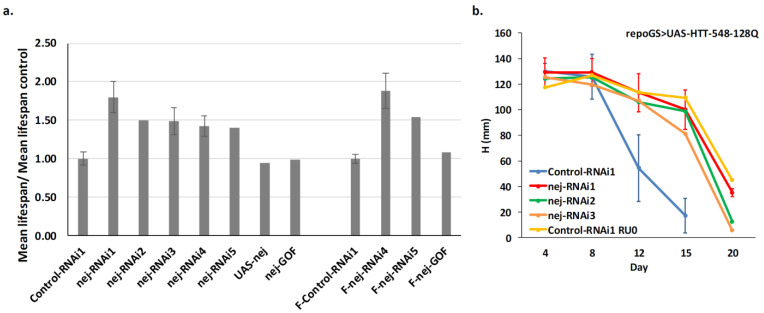
Extended longevity and improved locomotor activity of HD flies depleted for dCBP in glia. (**a**) Several independent RNAi transgenes targeting dCBP and transgenes leading to dCBP overexpression (see Appendix A for detailed genotypes) were assayed for their ability to modulate the lifespan of repoGS > UAS-HTT-548-128Q flies. All the RNAi lines tested increased the fly lifespan while the overexpressing lines have no effect, irrespective of the sex of the tested animals (F-: female flies, otherwise males). Error bars: standard deviation, provided when several independent experiments were performed. (**b**) The same flies were submitted to a negative geotaxis assay and their performance evaluated as the mean height at T = 7s. All the flies performed similarly until day 8 but, from day 12, the repoGS > UAS-HTT-548-128Q flies show a strong deficit of performance compared to control flies (*p* value < 0.0002, two sided *t*-test) which is almost abolished when dCBP is depleted. At 12 days, all the scores of the RNAi lines were not significantly different from the control line (*p* > 0.1, two sided *t*-test). At 15 days, only the score of the RNAi1 line was not significantly different from the control line (*p* < 0.05, two sided *t*-test) while the score of the RNAi2 and RNA3 lines were significantly different from the control line (*p* < 0.005, two sided *t*-test), reflecting incomplete rescue. Error bars: SE for N independent experiments (N = 2, 2, 5, 6, 4 at day 4, 8, 12, 15, 20, respectively).

**Figure 7 ijms-22-03884-f007:**
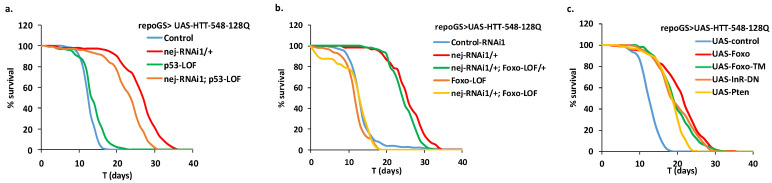
Extended longevity provided by glial depletion of dCBP is Foxo-dependent. We challenged repoGS > UAS-HTT-548-128Q flies exposed to RU100 to different genetic conditions (see Appendix A for detailed genotypes) and analyzed their lifespan. (**a**) Flies devoid of p53 (p53-LOF) are still strongly responsive to dCBP depletion (nej-RNAi1; p53-LOF) with lifespan increased by 83% compared to controls (MLS = 21.8 vs. 11.9 for controls; *n* = 100, 205, respectively; pLogRank < 10^−10^). (**b**) In contrast, flies devoid of Foxo (Foxo-LOF) are unable to respond to dCBP depletion (nej-RNAi1/+; Foxo-LOF) with lifespan similar to controls (MLS = 11.1 vs. 12.9 for controls; *n* = 107, 180, respectively; pLogRank > 0.2). When only one allele of *foxo* is lost (nej-RNAi1/+; Foxo-LOF/+), flies are still responsive (MLS = 23.9 vs. 12.9 for controls; *n* = 104, 180, respectively; pLogRank < 10^−10^). (**c**) Overexpression of wild type Foxo (UAS-Foxo), activated Foxo (UAS-Foxo-TM) or repression of insulin pathway with overexpression of a dominant negative insulin receptor (UAS-InR-DN) or the PTEN phosphatase (UAS-Pten) extend lifespan in repoGS > US-HTT-548-128Q flies. (MLS = 20.1, 18.9, 18.8, 17.4; *n* = 202, 170, 210, 206; pLogRank < 10^−10^, respectively). Control flies: MLS = 11.9, *n* = 226.

**Figure 8 ijms-22-03884-f008:**
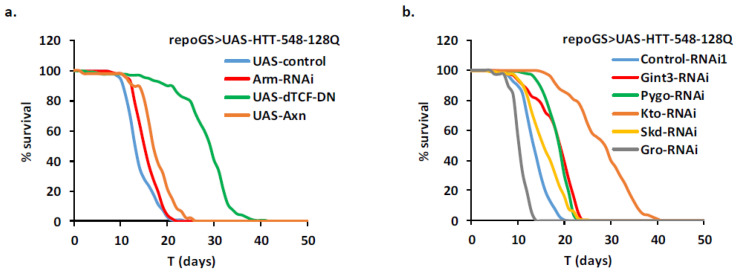
Identification of protective genetic interventions in the Wnt pathway. A repoGS > UAS-HTT-548-128Q line was crossed with transgenic lines allowing modulation of the Wnt pathway and the progeny (see Appendix A for detailed genotypes) was scored for their lifespan after induction with RU100. (**a**) Like in neurons, repression of the Wnt pathway in glial cells by depletion of arm (Arm-RNAi) and overexpression of a dominant negative *dTCF* gene (UAS-dTCF-DN) or of the *Axin* gene (UAS-Axn) are protective for mHTT toxicity (MLS = 27.4, 17.4; *n* = 142, 47; pLogRank < 10^−2^, 10^−10^, 10^−6^, respectively). Control flies: MLS = 13.3, *n* = 106. Note that repression of the Wnt pathway in glial cells by depletion of arm (Arm-RNAi) increased lifespan by only 11% (pLogRank < 10^−2^), below our 20% threshold criterium. (**b**) Additional members of the pathway (Gint3, pygo, kto) also protected the flies from glial mHTT toxicity when depleted (MLS = 17.5, 17.9, 27.4, 15.2; *n* = 202, 170, 210; pLogRank < 10^−10^, 10^−10^, 10^−10^, respectively). Note that depletion of skd (Skd-RNAi) increased lifespan by 17% (pLogRank < 10^−8^), below our 20% threshold criterium. In contrast, depletion of the groucho/TLE repressor (Gro-RNAi) reduced lifespan (MLS = 9.8; pLogRank < 10^−10^; *n* = 37). Control flies: MLS = 13.0, *n* = 126.

## Data Availability

Not applicable.

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
