# Peer review of "Genetic Screen in Adult Drosophila Reveals That dCBP Depletion in Glial Cells Mitigates Huntington Disease Pathology through a Foxo-Dependent Pathway"

_ijms, 2021, doi:10.3390/ijms22083884_

Round 1
Reviewer 1 Report
In the manuscript by Martin et al. the authors use a Drosophila disease model to investigate the contribution of glia cells to Huntington’s disease. Drosophila is successfully used to model HD for more than two decades, the authors use the inducible GeneSwitch system to drive mutant Huntingtin in adult glia only. The authors tested an enormous number of candidate genes for their potential effects on mutant Htt induced phenotypes (lifespan, motor abilities) and (1) identified several factors that affected lifespan in this glial model (e.g. PGC1a, regulators of cytoplasmic Ca2+ levels, CBP, Foxo, Wnt signaling) and (2) showed that the effects of some of these factors on pathology depends on the cell type (i.e. different from what was observed in neuronal models previously). The findings of the study are interesting and would advance the scientific field. However, some critical controls are missing and have to be done and the statistical analysis of the data needs revisiting also.
Controls:
- On Fig1a. the authors present survival curves of RU486 fed and not-fed repoGS>UAS-Htt flies. To demonstrate that the observed reduction in lifespan is due to Htt expression not the toxic effect of RU486, the authors should include the survival curve of RU486 (at 100 ug/ml) fed repoGS control flies (without UAS-Htt or better yet, with a UAS-Htt construct with a normal length polyQ stretch). I did notice that the effect (no-effect, actually) of RU486 on lifespan was presented on Fig. S1 but disturbingly, the median lifespan of control flies on Fig. S1 is approximately 15 days shorter than the uninduced repoGS>Htt RU0 control on Fig. 1A. These lifespan experiments (RU100 with Htt, RU100 without Htt, RU0 with Htt) should be repeated together in same experimental setup and the survival curves presented on the same plot.
- None of the RNAi or overexpressing lines are functionally validated in the study. As we do not really know which of these lines actually have a measurable effect on the mRNA levels of the targeted genes, negative phenotypical data cannot be really interpreted. The level of RNAi silencing or gene overexpression should be demonstrated and validated by qPCR (using universal GAL4 drivers) at least in the case of those genes, which are discussed in the main text and Figures.
- In their experiments the authors used carefully chosen UAS control constructs along with the experimental UAS-RNAi or UAS-OE transgenes. However, in none of the shown experiments was it demonstrated by direct measurements that the presence of the introduced UAS constructs does not lead to the simultaneous downregulation of UAS-Htt. As transcriptional downregulation of UAS-Htt leads to mitigated phenotypes, it is absolutely imperative to show that it is not the case. The authors did measure Htt protein levels by WB for their nejire experiments but since mutant Htt is an aggregating protein, visualizing the level of soluble Htt on a blot does not necessary reflect total protein load (and as a minor issue, the blots are not quantitated). The level UAS-Htt expression should be measured compared to the controls by qPCR (using repoGS drivers) at least in those experiments, which are discussed in the main text and Figures.
Statistics:
- The log-rank test is not adequate for testing mortality rates if lines on the survival plot are intertwined (there is both an increase and a decrease in mortality). This is the case on Fig. 2A, Fig. 3A, Fig. 4B, Fig. 7BC, and Fig. 8 B. Based on a paper by Li et al. [Li H, Han D, Hou Y, Chen H, Chen Z (2015) Statistical Inference Methods for Two Crossing Survival Curves: A Comparison of Methods. PLoS One. 2015; 10(1): e0116774.] I suggest to recalculate your survival/mortality data using Two Stage Hazard Rate Comparison described in [Qiu P, Sheng J. A two‐stage procedure for comparing hazard rate functions. Journal of the Royal Statistical Society: Series B (Statistical Methodology) 2008;70:191–208.]. One implementation of the method is the R package TSHRC.
- It is not clear how the authors choose 20% deviation from control in mean lifespan as a threshold, it seems arbitrary. Shouldn’t the threshold be based on a statistical significance value, or even better, a combination of significance and % deviation from control? In the interpretation of their data, the author sometimes rely on the 20% value, sometimes on P (for example on Fig. 8A Arm-RNAi is described as protective although it does not reach the 20% threshold, and on Fig. 8B there is a similar case with skd-RNAi). The authors should use a statistically valid threshold in a consistent manner.
- The data presented on Fig. 6A and Fig. 6B were not tested statistically. One-way ANOVA and Two-way ANOVA (or corresponding non-parametric tests) should be used, respectively.
Minor issues, typos:
L117: ImmobilonTM should read Immobilon™ (or without TM)
L157: orthologues provided
L231: succinate dehydrogenase
L236. Drosophila orthologue
L242: HD modifier genes
L245: “InsP3” should be spelled out at first use
L271: RNAi lines
L275: may result in the activation
L319: theres should be “control RNAi construct” instead of “control RNAi flies”
Fig. 7C. UAS-InR should be labeled as UAS-InR-DN (if this is a dominant negative construct as the main text suggests)
L467: short lifespan
L468: deacetylase that counteracts
L481: oxidative stress (OR) dependent manner
L484: suppressed in animals depleted for Foxo
L487: pathological effects
Reviewer 2 Report
The manuscript by Martin et al. reports the results from a genetic screen in flies aimed at identifying genetic modifiers of mutant Huntingtin (HTT-548-128Q) induced pathology in adult glial cells. They identify several modifiers of HD pathology that expand the number of therapeutic targets. Remarkably some modifiers act in glial cells but not in neuronal tissues. Depletion of dCBP acetyltransferase in glial cells strongly improved the phenotypes of mHTT expressing animals. They further demonstrate that the transcriptional activator Foxo is a downstream player in the glial protective pathway.
The results are intriguing and this manuscript could be acceptable for publication in Int J. Mol. Sci.
Major comments:
- The Western blot shown in Fig. S3 should be included in Figure 5. Moreover it is difficult to compare the loading control Lamin C with HTT. It does appear that the Lamin C levels are lower in the nej-RNAi3 line lane and higher in other lanes, making it difficult to determine whether HTT levels are down or similar to control RNAi. The authors need to provide quantification of multiple independent experiments, and statistical support for any decrease.
- The results in Figure 6 are not supported by statistical analysis. The authors should report in the figure legend: the statistical tests that have been used, the values (p values) and the differences that are statistically significant. It is not clear why in some cases the authors used females.
Minor
There are a few typos that should be corrected:
- Line 237: drosophila should be changed to Drosophila
- Figure S3 legend: targetetting should be changed to targeting
